# Most Custom Oral Appliances for Obstructive Sleep Apnea Do Not Meet the Definition of Custom

**DOI:** 10.3390/bioengineering12080798

**Published:** 2025-07-25

**Authors:** Leonard A. Liptak, Erin Mosca, Edward Sall, Shouresh Charkhandeh, Sung Kim, John E. Remmers

**Affiliations:** 1ProSomnus Sleep Technologies, Pleasanton, CA 94588, USA; emosca@prosomnus.com (E.M.); skim@prosomnus.com (S.K.);; 2Crouse Irving Memorial Sleep Lab, Go To Sleep Center, Scottsdale, AZ 85258, USA; 3Multidisciplinary Sleep Disorders Centre, Antwerp University Hospital, 2650 Edegem, Belgium; shouresh@gmail.com

**Keywords:** obstructive sleep apnea, sleep disordered breathing, respiratory medicine, snoring, oral appliance therapy, dental sleep medicine

## Abstract

Obstructive sleep apnea is a highly prevalent respiratory disease linked to increased morbidity and mortality, a reduced quality of life, and increased economic costs if not treated. Oral appliances are an emerging treatment option for obstructive sleep apnea. This review concluded that many oral appliances marketed as “custom” include modifications and prefabricated items, and therefore do not meet the definition of “custom” oral appliances. This misclassification could hinder the accurate characterization, evaluation, and appropriate prescription of oral appliances. To better inform the clinical utilization of custom oral appliances and to more closely align sleep medicine with the benefits of personalized medicine, we propose that the custom oral appliance classification be further refined into semi-custom and precision-custom categories.

## 1. Introduction

Obstructive sleep apnea (“OSA”) is a highly prevalent chronic respiratory disease that affects an estimated 1 billion people globally and 74 million in the United States [1,2]. Unmanaged OSA is linked to increased morbidity and mortality, a reduced quality of life, and significant economic costs [3].

Oral appliances (OAs), also known as mandibular advancement devices (“MADs”), mandibular repositioning devices (“MRDs”), or mandibular advancement splints (“MASs”), are an emerging treatment option for OSA, largely driven by the low patient compliance with continuous positive airway pressure (“CPAP”) therapy [4,5]. The clinical practice guidelines (“CPGs”) recommend custom, titratable OAs for patients who refuse or discontinue CPAP therapy or prefer an alternative [6]. Recently, another CPG recommended custom, titratable OAs as a frontline therapy for mild or moderate OSA and for severe OSA patients who fail or refuse CPAP trials [7].

This review evaluated whether the definition of “custom” from the AASM CPG effectively describes, classifies, and differentiates OAs for the treatment of patients with OSA. A more exact classification would inform the more appropriate clinical utilization of this therapy. The American Academy of Sleep Medicine (“AASM”)’s CPG for oral appliance therapy (“OAT”) defines custom OAs as devices “fabricated using digital or physical impressions and models of an individual patient’s oral structures” and “not primarily prefabricated items that are trimmed, bent, relined, or otherwise modified.” Prefabricated items are those manufactured without a specific patient in mind. “Records” refer to digital or physical dental casts that are used to create a custom OA [5].

## 2. Background

OAs have three components. The first is an overlay of the maxillary dental arch. The second is an overlay of the mandibular dental arch. The third is the titration mechanism.

Dental arch overlays are fabricated from the records of each individual patient’s oral structures. Records can be either digital scans or physical molds that are used to create dental casts. A proper fit between the overlays and the dental arches is crucial for mitigating the risk of dental side effects. Repositioning the mandible introduces orthodontic forces that can cause side effects. An overlay fit is essential for absorbing, distributing, and mitigating potentially injurious orthodontic forces. Prior research has identified a dose-dependent relationship between the degree of mandibular protrusion and the prevalence of dental side effects [8]. Overlays that poorly fit the dentition, use soft liner materials, or use other prefabricated dental retention items such as ball clasps may compromise the fit and increase the risks of dental side effects.

The third component of an OA is the titration mechanism. The titration mechanism articulates and stabilizes the mandibular overlay relative to the maxillary overlay to beneficially reposition the mandible and dilate the airway. Prior research has reported a dose-dependent association between stepwise mandibular repositioning and reductions in airway collapse events [9]. The target therapeutic mandibular location is determined by the healthcare provider. A bite registration is made at the target therapeutic mandibular location and is included in the record of oral structures provided to the OA manufacturer.

Overlays can be prefabricated or custom-made. Prefabricated overlays, often called “boil and bite” or “do-it-yourself” OAs, are considered non-custom OAs as per CPG definitions. Similarly, titration mechanisms can be prefabricated or custom. During our evaluation, we identified three categories of titration mechanisms: 1. prefabricated displacement screws; 2. prefabricated interchangeable connectors; and 3. custom, interlocking overlays.

## 3. Prefabricated Displacement Screws

Displacement screws are prefabricated items. They require overlay modifications to embed the prefabricated fixtures that anchor the displacement screws to the overlays. Examples include anterior-pull OAs and interlocking dorsal OAs with prefabricated screws, which can be seen in Figure 1 and Figure 2.

## 4. Prefabricated Connectors

Connector mechanisms are prefabricated items of various materials and sizes. The maxillary and mandibular overlays must be modified to embed the prefabricated fixtures that anchor the connector items to the overlays. Examples include push OAs and pull OAs with prefabricated connectors or Herbst arms. Figure 3, Figure 4 and Figure 5 provide examples of OAs with a prefabricated interchangeable connector and prefabricated fixture items.

## 5. Custom Interlocking Overlays

Each overlay has a monolithically embedded interlocking titration post that is directly designed based on the oral structures for each individual patient using computer-aided design and forward engineering technologies. There are no prefabricated items. No modifications are required. Each interlocking overlay is purely custom and has a specific mandibular advancement setting defined by the healthcare provider. The combination of different maxillary and mandibular overlays included within a treatment kit enables a range of specific protrusive or retrusive mandibular positions. Figure 6 provides an example of a dual-post OA with custom interchangeable overlays.

We hypothesized that most OAs labeled as custom do not meet the definition of custom. Most OAs marketed as custom use prefabricated titration mechanisms and are modified to embed these prefabricated mechanisms.

We propose two classifications: semi-custom and precision-custom OAs.

Semi-Custom Oral Appliances

Semi-custom OAs are made from the records of oral structures for each individual patient. Semi-custom OAs are either modified or they include prefabricated items, or both.

Precision-Custom Oral Appliances

Precision-custom OAs are directly and fully designed and manufactured based on the records of an individual patient’s oral records. They are not modified. They do not use prefabricated items. See Table 1 for a summary of our proposed OA classifications.

Further, we postulate that semi-custom OAs, characterized by prefabricated items and overlay modifications, are inherently more complex, which increases the tolerance stacks and may lead to reports of inconsistent performance.

## 6. Review

To challenge our proposed OA classifications, we evaluated OAT devices from two recent review articles [10,11], including 49 source articles referencing 74 OAT devices. Figure 7 provides a visual representation of our research approach. Of these, 51 were described as titratable and custom, with 41 providing enough detail for our evaluation. We assessed each of these devices based on the AASM CPG definition criteria for a custom oral appliance.

The 2015 AASM CPG’s definition for custom OAT devices can be distilled into two criteria. These two criteria are as follows:Is the OA made from records of an individual patient’s oral structures?Is the OA modified (trimmed, bent, relined) or primarily a prefabricated item?

The forty-one OAT devices from our review strategy were evaluated according to these two criteria.

## 7. Results


**Criterion 1: Made from records of an individual patient’s oral structures?**


All 41 OAs included in our analysis were made from individual patient records, based on descriptions contained in the source articles for each OA.


**Criterion 2: Is the OA modified or primarily a prefabricated item?**


A total of 32 of the 41, or 78%, of the OAs included in our analysis used prefabricated items. The prefabricated items used were titration mechanisms, anchor fixtures for the titration mechanism, or prefabricate retention items such as ball clasps. Each included overlay modifications to embed the prefabricated fixtures that anchor prefabricated titration mechanisms.

Only nine of the forty-one OAs fully met the two criteria of the AASM CPG’s definition for a custom OA. These nine did not use prefabricated items. They were not modified.

See Table 2 for an evaluation of each OA according to the definitional criteria.

## 8. Discussion

Our review and follow-up analysis confirmed our hypothesis that most OAs marketed as custom do not meet the AASM CPG’s criteria for custom OAs. Although 100% of the OAs in this analysis were made from oral records for an individual patient, 78% included at least one prefabricated component, or were modified, or both.

The fact that 32 of the 41 OAs used prefabricated titration mechanism items is potentially significant. It reveals that, for these 32 OAs, the titration mechanism—the component that most directly establishes and stabilizes the therapeutic mandibular reposition—is not made based on the oral records of each individual patient. It is prefabricated. It is generic.

Moreover, many of these prefabricated items are repurposed for the treatment of OSA. They were originally designed for other therapies. For example, displacement screws were designed for orthopedic procedures. Herbst arms were designed to correct Class II malocclusions in the field of orthodontics. These prefabricated items were not designed for the treatment of OSA, let alone custom treatment based on records of the oral structures of each individual patient.

Prefabricated titration mechanisms, which are not made based on individual patient records, limit customization, increase the complexity, and decrease consistency. Complex product designs, with more parts, modifications, and steps, are generally associated with a relatively lower quality and performance than less complex designs [46]. Occam’s razor, the principle of parsimony, applies: all things being equal, a simple product design will be better than a relatively more complex design.

This applies to OAs. Precision-custom OAs, with fewer parts, fewer steps, and no modifications, are inherently less complex and have relatively smaller tolerance stacks. Smaller tolerance stacks are associated with less variability and a more consistent, predictable performance. An inconsistent performance is often cited as a key barrier to increased OA utilization [47].

Personalized medicine, also known as precision medicine, optimizes therapies by matching treatments with individual patient characteristics [48]. Precision medicine is associated with an improved treatment efficacy [49], reduced adverse events [50], enhanced patient engagement [51], and cost-effectiveness [52].

Precision-custom OAs, by definition, are a step closer to the promise of personalized medicine than semi-custom OAs. Unlike semi-custom OAs, precision-custom OAs are exclusively made from the records of each individual patient’s oral records without the modifications and deviations necessary to accommodate prefabricated items. Modifications and prefabricated items make an OA more generic and less personalized.

### 8.1. Implications for Efficacy?

Studies have reported a dose-dependent relationship between mandibular advancement and therapeutic efficacy [9]. These investigations associate 2 mm stepwise increases in mandibular repositioning with clinically meaningful reductions in airway collapse events. Semi-custom OAs with larger tolerance stacks may result in clinically meaningful variances in mandibular positions. Studies have suggested that precision-custom OAs may offer different outcomes than semi-custom oral appliances [10,37,39]. An in vitro study that measured mandibular repositioning errors reported that OAs meeting the requirements of the proposed precision-custom category demonstrated an average error of just 0.32 mm, in contrast to an average mandibular repositioning error of 3.74 mm for OAs that meet the requirements of the proposed semi-custom category (with prefabricated components and/or modifications) [53]. In other words, the average mandibular repositioning error for these semi-custom OAs exceeded the 2 mm threshold associated with changes in efficacy, while the average error for the precision-custom category of devices did not, suggesting a more consistent, predictable ability to reposition and stabilize the mandible at the prescribed mandibular position.

### 8.2. Signs of Different Outcomes Between Semi-Custom and Precision-Custom OAs

It takes time to investigate new technologies. For example, the 53 studies referenced by the 2015 AASM CPG are, on average, over 18 years old [6]. However, are there any recent records or signs that associate precision-custom OAs with different results?

#### 8.2.1. Signs of a Different Efficacy

A prospective study prescribed precision-custom OAs for 288 patients with a 50%, 31%, and 19% mix of mild, moderate, and severe OSA. The study reported 85% success treating these patients to an AHI < 10, and 73% treating patients to an AHI < 5 [45]. A second prospective study prescribed precision-custom OAs to treat 48 patients with a mix of mild, moderate, and severe OSA. The study reported 88% success treating patients to an AHI < 10 and a 50% improvement over the baseline [44]. Although not comparative randomized controlled studies, these results represent rates of efficacy that are directionally different than what has been previously reported for semi-custom OAs [11].

#### 8.2.2. Signs of Different Patient Preferences

One randomized controlled cross-over trial compared the patient preference between precision-custom OAs and semi-custom OAs [54]. Of the ten patients who completed the study, eight preferred the precision-custom OA. These patients cited comfort, ease of use, and durability as reasons for their preference. The two patients who preferred the semi-custom OA cited ease of use in combination with their CPAP.

#### 8.2.3. Signs of Different Symptom Alleviation

A prospective, comparative, randomized controlled trial reported a 91% success rate of treating snoring with a precision-custom OA, in comparison with 58% for combined airway and positional therapy [55]. In a different study, a single-arm investigation that included a precision-custom OA for treatment, 85% of the patients stated that they achieved their treatment goals, and 97% reported a reduction in snoring [43].

#### 8.2.4. Signs of Different Side Effects

A single-arm, prospective, longitudinal study involving the treatment of OSA patients with precision-custom OAs reported no clinically or statistically significant changes in tooth position, overbite, or overjet after a two-year follow-up period [56]. A second, single-arm, comparative, longitudinal study evaluated the side effects associated with treating OSA with precision-custom OAs. The study also reported no clinically meaningful changes in the orthodontic conditions after a two-year follow-up period [57]. A third study more broadly reported the side effects of using precision-custom OAs to treat OSA patients. This study reported no serious adverse events and no adverse events that resulted in the discontinuation of therapy [43]. Again, these are not randomized controlled trials, but they do indicate results that are different from what is commonly reported for OAT [6].

#### 8.2.5. Signs of Different Adverse Event Reports

A review of the US Food and Drug Administration (“FDA”)’s Manufacturer and User Device Experience (“MAUDE”) indicates a difference in the adverse event reports associated with semi-custom and precision-custom OAs. The FDA’s MAUDE database contains reports of serious adverse events associated with FDA-cleared medical devices. There were 853 adverse event reports over the past ten years filed in the product categories allocated for OAs used in the treatment of OSA [58]. Of these 853 reports, 86% of the adverse events were associated with semi-custom OAs, while 13% of the adverse event reports involved non-custom OAs. In contrast, 1% of the adverse event reports involved precision-custom OAs. Semi-custom, non-custom, and precision-custom classification determinations were made based on publicly available device labeling.

A difference in the cost–benefit ratio is one potential implication of the MAUDE database finding, and another reason to consider the semi-custom and precision-custom categories. Adverse events are expensive. They often disrupt treatment for the patient, require emergency interventions from the provider, and may ultimately result in the discontinuation of treatment. The higher quantum of adverse events linked with semi-custom devices suggests a less favorable cost–benefit ratio relative to precision-custom devices or even non-custom devices. Leaving semi-custom and precision-custom OAs bundled into the custom OA category would prevent providers, patients, and payers from discerning this important difference in costs and benefits.

It should be noted that the FDA MAUDE database has limitations, chiefly that it relies on patients, providers, or device manufacturers to self-report adverse events.

There are several additional limitations to this study. One is that it is based on primary source records that are inconsistent in their controls, their definitions, their mix of OSA severity in their patient populations, and other variables that might be relevant to our research topic. The limitations in the data structures of the source documents inhibit ad hoc analyses that could control for confounding factors such as the severity, BMI, gender, and age.

Another limitation is the inconsistent approach to OA titration utilized across the source documents. Titration refers to adjusting the titration mechanism of the OA to alter the anterior–posterior positioning of the mandible to dilate the airway. It is thought that subjective titration, typically based on patient-reported symptoms, is not a good predictor of the response to treatment, and it is thought that the OA efficacy may be improved when informed by objective testing tools such as oximetry [28]. However, most source documents fail to describe their approaches to titration. Several source studies have referenced the use of subjective feedback, such as symptoms, to inform titration. However, just one article disclosed the utilization of oximetry to inform the titration process [37].

A remedy for these limitations would be a prospective, randomized controlled investigation that directly compares the efficacy of semi-custom and precision-custom styles of oral appliances, with an emphasis on controlling potentially confounding variables.

## 9. Conclusions

Most OAs labeled as “custom” do not meet the definitional conditions of “custom”. This analysis reports that, for many OAs labeled as custom, the titration mechanism is primarily a prefabricated component that necessitates further modification to the overlay components. The inclusion of one or more prefabricated components and the need for modification(s) violate the established definition of a custom OA.

For this reason, the authors propose segmenting the custom OA category into semi-custom and precision-custom OAs. Semi-custom OAs are those that are custom-made from oral records but include prefabricated components and/or modifications. Precision-custom OAs are those that are 100% fabricated based on the records of each patient’s oral structures, without any prefabricated components or modifications.

Prior research and follow-up analyses have indicated differences in the performance between semi-custom and precision-custom types of OAs. Although not directly tested in head-to-head, randomized controlled studies with large sample sizes, there are initial signs of differences in the efficacy, accuracy of mandibular repositioning and stabilization, side effects, and adverse events. Additionally, there are head-to-head studies, albeit with limited scopes and sample sizes, that demonstrate significant differences between precision treatments and semi-custom OAs with respect to patient preferences, nightly utilization, symptoms such as snoring, and adverse events.

Further, established device-engineering principals of medical devices and products, such as design elegance, complexity factors, and tolerance stacks, aptly explain how an OA that uses prefabricated items and requires modifications could be associated with a less consistent, predictable performance; a lower quality; and a reduced durability relative to an OA that does not, thus impacting the cost–benefit calculus of the therapy.

In closing, we propose that the custom OA classification be further refined into semi-custom and precision-custom categories to better inform clinical use and future research efforts.

## Figures and Tables

**Figure 1 bioengineering-12-00798-f001:**
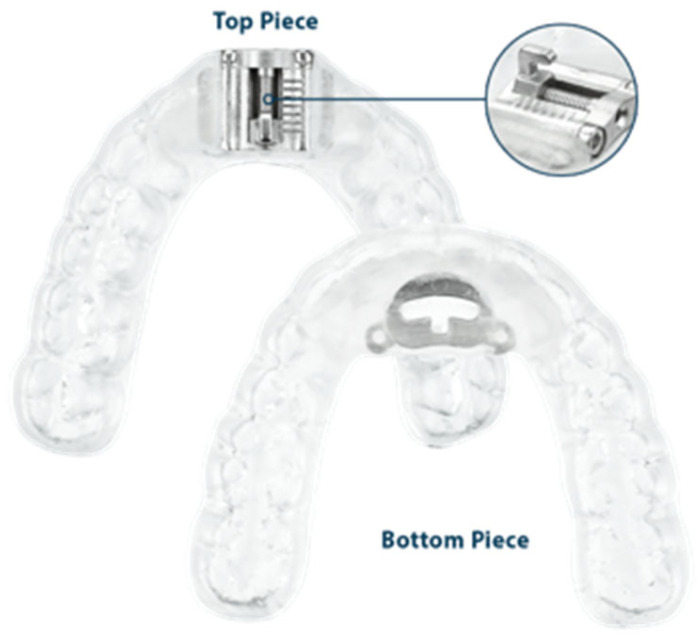
Anterior-pull OA with prefabricated displacement screws.

**Figure 2 bioengineering-12-00798-f002:**
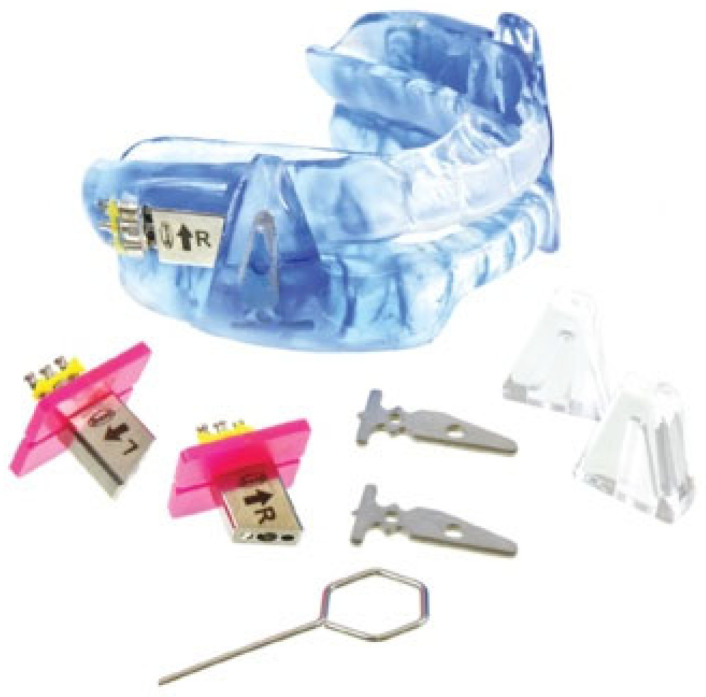
Interlocking dorsal OA with prefabricated screws.

**Figure 3 bioengineering-12-00798-f003:**
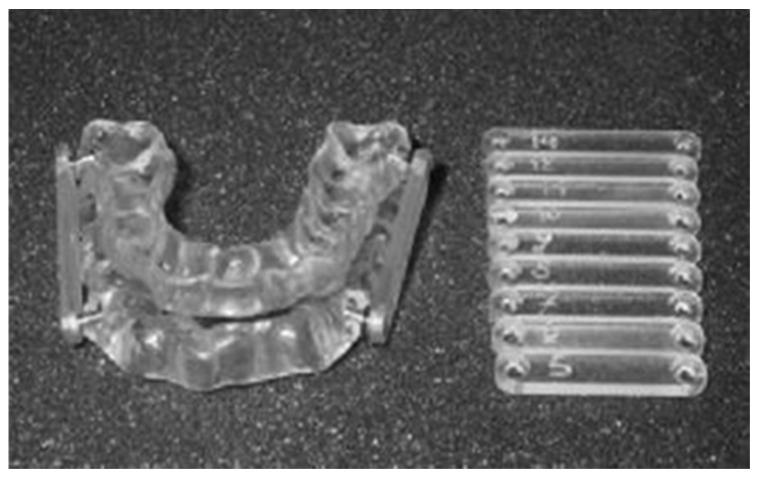
Push OA with prefabricated connectors of different lengths.

**Figure 4 bioengineering-12-00798-f004:**
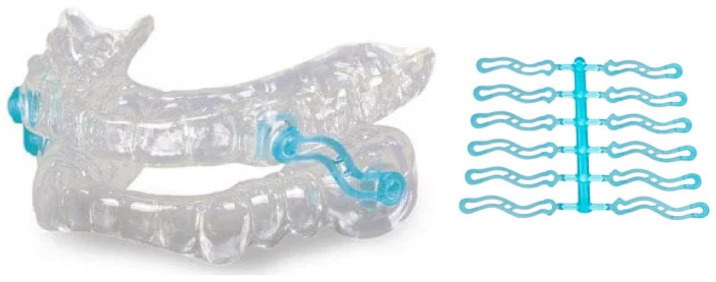
Pull OA with prefabricated connectors of different lengths.

**Figure 5 bioengineering-12-00798-f005:**
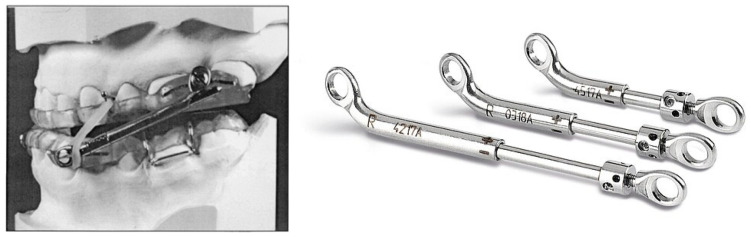
Push OA with prefabricated Herbst arms of different lengths.

**Figure 6 bioengineering-12-00798-f006:**
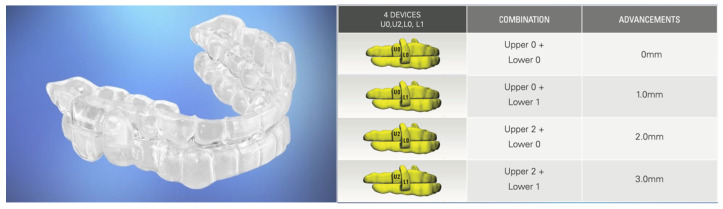
Dual-post OA with interchangeable overlay components.

**Figure 7 bioengineering-12-00798-f007:**
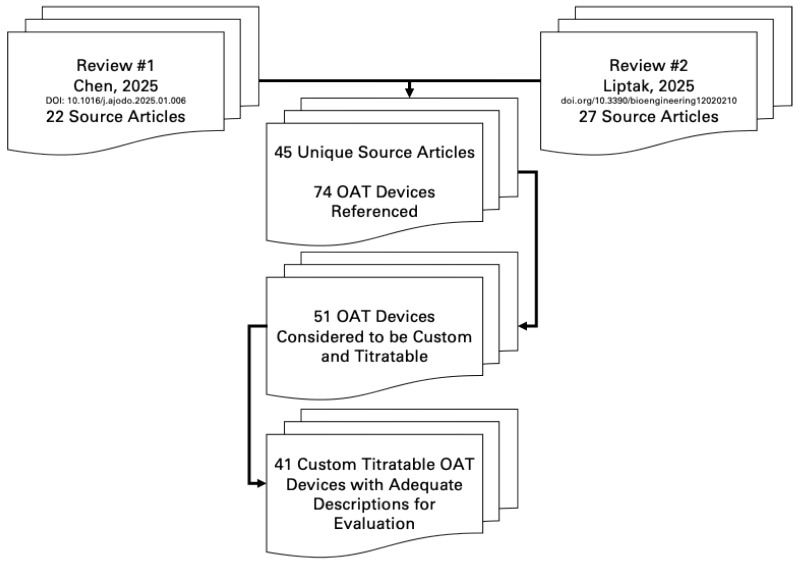
Review approach [10,11].

**Table 1 bioengineering-12-00798-t001:** Proposed OA classifications.

	Custom OA Definition Criteria (Summarized)		
Proposed Classification	Made from Oral Records of an Individual Patient?	Modified (Trimmed, Bent, Relined)?	And/Or IncludesPrefabricated Items?
Semi-Custom	Yes	Yes	
Precision-Custom	Yes	No	

**Table 2 bioengineering-12-00798-t002:** Evaluation of OAs.

Review Article	Author	Reference	Basic Description	1. Made from Records of Oral Structures?	2. Is the Device Modified or Primarily Prefabricated?
Chen 2025	Bloch 2000	[12]	Push	Yes	Yes
Chen 2025	Pepin 2019	[13]	Push	Yes	Yes
Chen 2025	Pepin 2019	[13]	Push	Yes	Yes
Liptak 2025	Randerath 2002	[14]	Push	Yes	Yes
Liptak 2025	Ghazal 2008	[15]	Push	Yes	Yes
Liptak 2025	Gagnadoux 2009	[16]	Push	Yes	Yes
Chen 2025	Yanamoto 2021	[17]	Pull	Yes	Yes
Chen 2025	Zhou 2012	[18]	Pull	Yes	Yes
Chen 2025	Isacsson 2017	[19]	Pull	Yes	Yes
Liptak 2025	Vecchierini 2016	[20]	Pull	Yes	Yes
Liptak 2025	Vecchierini 2016	[20]	Pull	Yes	Yes
Liptak 2025	Tegelberg 2020	[21]	Pull	Yes	Yes
Liptak 2025	Kuna 2006	[22]	Pull	Yes	Yes
Liptak 2025	Henke 2000	[23]	Pull	Yes	Yes
Liptak 2025	Vanderveken 2024	[24]	Pull	Yes	Yes
Liptak 2025	Schneiderman 2020	[25]	Pull	Yes	Yes
Liptak 2025	Pancer 1999	[26]	Pull	Yes	Yes
Liptak 2025	Ghazal 2009	[15]	Pull	Yes	Yes
Chen 2025	Gagnadoux 2009	[16]	Interlocking	Yes	Yes
Liptak 2025	Vanderveken 2024	[24]	Interlocking	Yes	Yes
Liptak 2025	Van Haesendonck 2016	[27]	Interlocking	Yes	Yes
Liptak 2025	Schneiderman 2020	[25]	Interlocking	Yes	Yes
Liptak 2025	Remmers 2013	[28]	Interlocking	Yes	Yes
Liptak 2025	Mehta 2001	[29]	Interlocking	Yes	Yes
Liptak 2025	de Ruiter 2020	[8]	Interlocking	Yes	Yes
Chen 2025	Abd-Ellah 2024	[30]	Bi-Block	Yes	Yes
Chen 2025	Isacsson 2019	[31]	Bi-Block	Yes	Yes
Chen 2025	Bosschieter 2022	[32]	Anterior	Yes	Yes
Chen 2025	Johal 2017	[33]	Anterior	Yes	Yes
Chen 2025	Friedman 2012	[34]	Anterior	Yes	Yes
Chen 2025	Lettieri 2011	[35]	Anterior	Yes	Yes
Chen 2025	Sari 2011	[36]	Anterior	Yes	Yes
Liptak 2025	Stern 2021	[37]	Dual-Post	Yes	No
Liptak 2025	Silva 2024	[38]	Dual-Post	Yes	No
Liptak 2025	Sall 2023	[39]	Dual-Post	Yes	No
Liptak 2025	Sall 2021	[40]	Dual-Post	Yes	No
Liptak 2025	Remmers 2017	[41]	Dual-Post	Yes	No
Liptak 2025	Murphy 2021	[42]	Dual-Post	Yes	No
Liptak 2025	Mosca 2022	[43]	Dual-Post	Yes	No
Liptak 2025	Kang 2022	[44]	Dual-Post	Yes	No
Liptak 2025	Knowles 2021	[45]	Dual-Post	Yes	No

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
