# Peer review of "Most Custom Oral Appliances for Obstructive Sleep Apnea Do Not Meet the Definition of Custom"

_bioengineering, 2025, doi:10.3390/bioengineering12080798_

Round 1
Reviewer 1 Report
Comments and Suggestions for Authors
Dear Editor,
I would like to thank the authors of the manuscript bioengineering-3708251 entitled "Most Custom Oral Appliances for Obstructive Sleep Apnea Do Not Meet the Definition of Custom “for presenting the results of their review study on the effect of misclassifying oral appliances for OSA on the quality of OSA treatment and patient satisfaction.
The authors used a definition of custom oral appliances (OA) adopted from the AASM clinical practice guidelines for Oral Appliance Therapy (“OAT”) further focusing on the two basic elements of the definition:
- fabricated using digital or physical impressions and models of an individual patient’s oral structures
2.not primarily prefabricated items that are trimmed, bent, relined, or otherwise modified."
After reading this article in details, my main impression is that the article is very well written, in adherence to Journal's standards, and addresses a very important issue of the effect of prefabricated elements in OAT for OSA on the quality of OSA treatment.
The authors gave a detailed explanation on the elements of an OA, with the focus on titration mechanisms, where most OA manufacturers still used prefabricated elements. The authors suggested that use of these elements is clearly in contrast to the OA definition criterion 2. In the review of the current literature, the authors based their findings on two recently published review papers, one of which is produced by the same research group. . They stated that, by their opinion use of some prefabricated elements in a custom made OA is in contrast to the definition in AASM CPG and could have impact on quality of care. The authors further proposed reclassification of so called custom OA in two groups: semi –custom and precision custom devices.
In the discussion segment, the authors provided the references to studies in favor of precision custom devices, but do not provide a lot of comparative data on so called semi-custom OA. There is no data presented on real difference in efficacy between so called semi-custom vs precision custom OA that would justify this new classification. Large studies have even shown minimal differences between prefabricated vs custom made devices concerning AHI and ODI lowering, with differences mostly in patient comfort and adherence.
Most of the reference studies presented focus on one producer of precision devices, while the authors failed to mention details of the studies, such as applicability in military personnel settings, or the small number of subjects, or lack of control groups using semi-custom devices. Some of the references quote the studies focused on the effect of OA devices on bruxism or snoring, not OSA, or compare precision OA devices with other forms of therapy. Again, on analyzing the side effects, the authors focus on precision made OA, missing the comparison to semi-custom OA (line 219, reference missing).
The authors need to provide better discussion with comparison of so called precision custom devices with semi-custom devices in order to justify this new classification. Prefabrication of some elements of titration mechanisms does not immediately imply that this is not a custom made OA, so suggesting this new classification needs more justification.
Reviewer 2 Report
Comments and Suggestions for Authors
Oral appliances offer a potentially more convenient and less obtrusive alternative to continuous positive airway pressure (CPAP) therapy for patients experiencing obstructive sleep apnea, especially in cases with mild to moderate obstructive sleep apnea.
This review presents well-reasoned arguments refining the classification of custom oral appliances. It highlights the current limitations of oral appliances such as reliance on prefabricated components (such as displacement screws, and interchangeable connectors).
Review emphasizes the importance of design complexity and tolerance stacks, raising important questions concerning the performance and efficiency of different types of oral appliances types and its customization. Proposed refinement of the custom classification of oral appliances is clearly practical and clinically relevant recommendation.
I believe this review can be recommended for publication.
Some minor comments:
-
In the Discussion, the authors explore potential benefits of precision-custom oral appliances, presenting evidence suggestive of advantages over semi-custom devices across key areas: efficacy, patient preference, symptom alleviation, and side effects. While these ‘signs’ are promising, it’s crucial to consider the potential influence of confounding factors. Specifically, to what extent might the observed differences be modulated by the severity of a patient’s obstructive sleep apnea (e.g. depending on Apnea-Hypopnea Index scale), their age, and/or gender?
-
Given the potential benefits of precision-custom oral appliances highlighted in this study, and recognizing the increasing utilization of tele-health and remote monitoring strategies in healthcare, it would be valuable to explore whether the authors have identified any research that investigates the relationship between oral appliances customization level and the efficacy of adjunctive home-confinement measures, such as home oximetry. Specifically, are there any studies that compare the utility of home oximetry in patients using precision-custom versus semi-custom oral appliances,? Understanding whether a more customized appliance influences the reliability or interpretability of data obtained through home oximetry could have implications for remote monitoring strategies and the long-term management of obstructive sleep apnea, particularly in situations where in-person assessments are limited
-
P.8., L.217: In this sentence: “Again, these are not randomized controlled trials, but they do indicate results that are different from what is commonly reported for OAT (..)” supposed citation is missing. Please provide.
Reviewer 3 Report
Comments and Suggestions for Authors
I thank the authors for the opportunity to read this interesting paper.
I believe it is an interesting review of custom oral appliances for obstructive sleep apnea and may be useful to readers interested in the topic.
I advise specifying the meaning of acronyms the first time they appear (e.g., AASM).
Round 2
Reviewer 1 Report
Comments and Suggestions for Authors
Dear Editor,
I would like to thank the authors of the revised manuscript bioengineering-3708251 entitled "Most Custom Oral Appliances for Obstructive Sleep Apnea Do Not Meet the Definition of Custom “for presenting the results of their review study on the effect of misclassifying oral appliances for OSA on the quality of OSA treatment and patient satisfaction.
The authors used a definition of custom oral appliances (OA) adopted from the AASM and AADSM clinical practice guidelines for Oral Appliance Therapy (“OAT”) further focusing on the two basic elements of the definition:
- fabricated using digital or physical impressions and models of an individual patient’s oral structures
2.not primarily prefabricated items that are trimmed, bent, relined, or otherwise modified."
After reading this article in details, my main impression is that the article is very well written, in adherence to Journal's standards, and addresses a very important issue of the effect of prefabricated elements in OAT for OSA on the quality of OSA treatment.
The authors gave a detailed explanation on the elements of an OA, with the focus on titration mechanisms, where most OA manufacturers used prefabricated elements. The authors suggested that use of these elements was clearly in contrast to the OA definition criterion 2. In the review of the current literature, the authors based their findings on two recently published review papers, one of which is produced by the same research group as the authors. They stated that, by their opinion, use of some prefabricated elements in a custom made OA is in contrast to the definition in AASM CPG and could have impact on quality of care. The authors further proposed reclassification of so called custom OA in two groups: semi –custom and precision custom devices.
In the discussion segment, the authors provided the references to studies in favor of so called precision custom devices, but do not provide a lot of comparative data on so called semi-custom OA. There is no data presented on real difference in efficacy between so called semi-custom vs precision custom OA that would justify this new classification. Large studies have even shown minimal differences between prefabricated vs custom made devices concerning AHI and ODI lowering, with differences mostly in patient comfort and adherence.
Most of the reference studies presented focused on one producer of precision devices, while the authors failed to mention details of the studies, such as applicability in military personnel settings, or the small number of subjects, or lack of control groups using semi-custom devices. Some of the references quote the studies focused on the effect of OA devices on bruxism or snoring, not OSA, or compare precision OA devices with other forms of therapy.
Analyzing the side effects, the authors introduced the data from FDA MAUDE base, declaring that only 1% of the OA complaints are made for precision made OA, in comparison to 86% related to semi- custom OA and only 13% for non-custom OA? This would imply that simple Boil and bite devices are the optimal solution, since they are the cheapest, OTC available and carry significantly less adverse effects than semi-custom devices, which is in contrast to previous claims in this review. Please discuss these findings.
The authors need to provide better discussion with comparison of so called precision custom devices with semi-custom devices in order to justify this new classification. Prefabrication of some elements of titration mechanisms does not immediately imply that this is not a custom made OA, so suggesting this new classification needs more justification.
Minor issues: references in superscript halfway down transform to references in brackets?
Round 3
Reviewer 1 Report
Comments and Suggestions for Authors
Dear Editor,
I would like to thank the authors of the revised manuscript bioengineering-3708251 entitled "Most Custom Oral Appliances for Obstructive Sleep Apnea Do Not Meet the Definition of Custom “for presenting the results of their review study on the effect of misclassifying oral appliances for OSA on the quality of OSA treatment and patient satisfaction.
The authors used a definition of custom oral appliances (OA) adopted from the AASM and AADSM clinical practice guidelines for Oral Appliance Therapy (“OAT”) further focusing on the two basic elements of the definition:
- fabricated using digital or physical impressions and models of an individual patient’s oral structures
2.not primarily prefabricated items that are trimmed, bent, relined, or otherwise modified."
After reading this article in details, my main impression is that the article is very well written, in adherence to Journal's standards, and addresses a very important issue of the effect of prefabricated elements in OAT for OSA on the quality of OSA treatment.
The authors gave a detailed explanation on the elements of an OA, with the focus on titration mechanisms, where most OA manufacturers used prefabricated elements. The authors suggested that use of these elements was clearly in contrast to the OA definition criterion 2. In the review of the current literature, the authors based their findings on two recently published review papers, one of which is produced by the same research group as the authors. They stated that, by their opinion, use of some prefabricated elements in a custom made OA is in contrast to the definition in AASM CPG and could have impact on quality of care. The authors further proposed reclassification of so called custom OA in two groups: semi –custom and precision custom devices.